# Improvement of classification performance of Parkinson's disease using shape features for machine learning on dopamine transporter single photon emission computed tomography

Takuro Shiiba[1]*, Yuki Arimura[2], Miku Nagano[3], Tenma Takahashi[3], Akihiro Takaki[1]

1 Department of Radiological Technology, Faculty of Fukuoka Medical Technology, Teikyo University, Misakimachi, Omuta-shi, Fukuoka, Japan, 2 Department of Radiology, Kokura Medical Center, Harugaoka, Kokura Minami-ku, Kitakyushu-shi, Fukuoka, Japan, 3 Department of Radiology, University of Miyazaki Hospital, Kihara, Kiyotake-cho, Miyazaki-shi, Miyazaki, Japan

* shiiba@fmt.teikyo-u.ac.jp

**Data Availability Statement:** The data underlying the results presented in the study are available

## Abstract

### Objective

To assess the classification performance between Parkinson's disease (PD) and normal control (NC) when semi-quantitative indicators and shape features obtained on dopamine transporter (DAT) single photon emission computed tomography (SPECT) are combined as a feature of machine learning (ML).

### Methods

A total of 100 cases of both PD and normal control (NC) from the Parkinson's Progression Markers Initiative database were evaluated. A summed image was generated and regions of interests were set to the left and right striata. Area, equivalent diameter, major axis length, minor axis length, perimeter and circularity were calculated as shape features. Striatum binding ratios (SBR$_{putamen}$ and SBR$_{caudate}$) were used as comparison features. The classification performance of the PD and NC groups according to receiver operating characteristic analysis of the shape features was compared in terms of SBRs. Furthermore, we compared the classification performance of ML when shape features or SBRs were used alone and in combination.

### Results

The shape features (except minor axis length) and SBRs indicated significant differences between the NC and PD groups ($p < 0.05$). The top five areas under the curves (AUC) were as follows: circularity (0.972), SBR$_{putamen}$ (0.972), major axis length (0.945), SBR$_{caudate}$ (0.928) and perimeter (0.896). When classification was done using ML, AUC was as follows: circularity and SBRs (0.995), circularity alone (0.990), and SBRs (0.973). The classification

from Parkinson's Progression Markers Initiative (PPMI)(https://www.ppmi-info.org).

**Funding:** This work received a grant from JSPS KAKENHI (Grant Number 18K15565) to TS. The funder had no role in study design, data collection and analysis, decision to publish, or preparation of the manuscript.

**Competing interests:** The authors have declared that no competing interests exist.

performance was significantly improved by combining SBRs and circularity than by SBRs alone (p = 0.018).

## Conclusion

We found that the circularity obtained from DAT-SPECT images could help in distinguishing NC and PD. Furthermore, the classification performance of ML was significantly improved using circularity in SBRs together.

## Introduction

Parkinson's disease (PD) is characterized by motor symptoms, such as tremor, muscular rigidity, immobility and postural reflex disorder, and involves frequent complications of non-motor symptoms, such as autonomic nervous disorder, depression, sleep disturbance and dementia [1]. The incidence of PD has increased by more than double over the past 26 years, from 2.5 million patients in 1990 to 6.1 million patients in 2016 [2]. Pathologically, PD is characterized by the degeneration of the nigrostriatal dopamine nerve and the appearance of inclusion bodies containing α-synuclein as the main component, i.e. Lewy body [3,4]. The striatum to which dopamine neurons are projected is one of the nerve nuclei constituting the basal ganglia and comprises the caudate nucleus and putamen. Dopamine transporter (DAT)-single photon emission computed tomography (SPECT) contributes to the diagnosis of PD and Lewy body dementia by providing a SPECT image reflecting DAT distribution density in the striatum. In general, the evaluation of DAT-SPECT images is visually performed using semi-quantitative indicators, such as specific binding ratio [5–8]. In visual assessment, information regarding the asymmetry of the left and right striata and accumulation site of $^{123}$I-FP-CIT can be obtained [9–12]. Conversely, semi-quantitative indicators can provide information regarding the count of the striatum in the background; however, the information of the striatum shape cannot be obtained. Few studies have used the shape of the striatum as a feature. Oliveira et al. [13] described that the length of the striatal uptake region revealed clinical added value because the accuracy obtained was slightly higher than the best accuracy achieved by the standard uptake ratio-based features. Staff et al.[14] indicated that the ratio of the long-to-short axis of the shape of the striatal uptake was as good as the putamen background ratio and experienced expert observers. Thus, the usefulness of using the shape feature in combination with the semi-quantitative indicator is evident. Further, it is well known that typical PD indicates egg or dot shape because a decrease in the uptake of the striatum occurs from the putamen and caudate uptake is retained. Kahraman et al. reported that 87 out of 120 cases of PD showed egg shape [9]. Therefore, we believed that it would be suitable to distinguish between PD and NC using the circularity of the striatal accumulation shape as a feature.

Machine learning is increasingly used in medical image identification and is also applied to the classification of DAT-SPECT image for the diagnosis of PD[15–19]. Generally, it is thought that the use of machine learning (ML) could improve the classification accuracy because discriminative features can be simultaneously used to build a more robust multidimensional classification model, as opposed to the models built based on a single feature. Also, in PD and NC classification using ML, the combination of semi-quantitative indicators and shape features could improve classification performance. The Development of an automatic DAT-SPECT diagnosis system that takes advantage of shape features can be divided into two parts. One is the extraction of the striatum. The other is the calculation and selection of effective shape features. We focused on calculation and selection of shape features.

This study aimed to indicate the usefulness of using circularity in shape features and assess the classification performance between PD and NC when semi-quantitative indicators and circularity are combined as a feature of ML.

## Materials and methods

### Parkinson's progression markers Initiative (PPMI) database

The mission of PPMI is to identify one or more biomarkers of PD progression, a critical next step in the development of new and better treatment for PD. PPMI establishes comprehensive, standardized, longitudinal PD data and biological sample repository that is available to the research community [20]. All data used in this study were obtained from the PPMI database (www.ppmi-info.org/data) available on April 3, 2018. The dataset contained all 625 pre-processed [123]I-FP-CIT SPECT brain images acquired at the screening stage. A total of 100 cases of both PD and normal control (NC) were randomly selected. The PD group included 60 men and 40 women (65.7 ± 9.9 years, age range: 31–84 years), and the NC group included 57 men and 43 women (59.8 ± 11.5 years, age range: 39–89 years). SPECT images of the burst striatum type [9,11,21] were not included in selected groups. The burst striatum type is severe bilateral reduction with almost no uptake in either the putamen or caudate[7].

Informed consents were obtained for clinical testing and neuroimaging from the participants of the PPMI cohort. The study was approved by the institutional review boards of all participating institutions. We declare that all procedures in this study have been performed in accordance with the ethical standards laid down in the 1964 Declaration of Helsinki and its later amendments.

### SPECT image processing and calculation of striatum binding ratio (SBR) by PPMI

Preprocessed SPECT images and SBRs were downloaded from the PPMI website. As by PPMI documentation, preprocessing steps were performed at the Institute for Neurodegenerative Disorders (IND, New Haven, CT) and included the following steps: SPECT imaging and reconstruction: SPECT imaging was acquired at each imaging centers as per the PPMI imaging protocol and sent to the institute for neurodegenerative disorders for processing and calculation of SBRs. SPECT raw projection data were imported to a HERMES (Hermes Medical Solutions, Stockholm, Sweden) system for iterative reconstruction. Iterative reconstruction was done without any filtering. The reconstructed files were transferred to the PMOD (PMOD Technologies, Zurich, Switzerland) for subsequent processing. Attenuation correction ellipses were drawn on the images and a Chang 0 attenuation correction was applied to images utilizing a site-specific μ that was empirically derived from phantom data acquired during site initiation for the trial. Once attenuation correction was completed, a standard Gaussian three-dimensional 6.0 mm filter was applied. Then, these files were normalized to standard Montreal Neurologic Institute space so that all scans were in the same anatomical alignment. The preprocessed images were saved as a DICOM format using $91 \times 109 \times 91$ cubic voxels with 2 mm.

The calculation method of SBR as performed at the IND was as follows: the transaxial slice with the highest striatal uptake was identified, and the eight hottest striatal slices around it were averaged to generate a single slice image. Regions of interests (ROIs) were placed on the left and right striatal ROIs were covering and including all activity visualised in putamen and caudate (target region), and the occipital cortex (reference region). Count densities for each region were extracted and used to calculate the SBRs for each of the four striatal regions (left

and right SBR$_{caudate}$, left and right SBR$_{putamen}$). SBRs were calculated as (target region/reference region)−1 [22].

## Calculation of image feature

SPECT image features were calculated using MATLAB2018a (The MathWorks, Inc. Massachusetts, USA). We thought that the error and bias would increase if the contrast between the striatum and the background was low in a single SPECT image for ROI settings. Thus, multiple images were summed. Preliminary experiments showed that the maximum value above the parotid gland was in the left or right striatum. First, the maximum value and position of each slice above the parotid gland were searched. Next, a slice with the maximum value of the striatal part was searched. Then, a summed image was generated from the slice with maximum value and plus or minus two slices from the upper and lower slices (summed range: 1 cm). Region of interests were set to the left and right striata of the summed image by a radiological technologist who has experience in nuclear medicine field for 10 years. The region where the radioactivity is visually accumulated at the site where the striatum exists anatomically was manually delineated. The calculated shape features were as follows: area, equivalent diameter, major axis length, minor axis length, perimeter, and circularity. The circularity was calculated by following equation;

$$circularity = \frac{4\pi S}{L^2},$$

here, S is the area of ROI, and L is the perimeter of ROI. Intensity features were maximum and minimum pixel count and the mean pixel count of the ROI.

## Classification using machine learning

We used support vector machine (SVM) as a classifier for the classification of PD and NC. The SVM binary classification algorithm searches for an optimal hyperplane that separates the data into two classes, e.g., PD and NC. For separable classes, the optimal hyperplane maximizes a margin surrounding itself, which creates boundaries for the positive and negative classes. The features were standardized before learning. Leave-one-out cross validation (LOOV) method was performed to improve generalization performance. We compared classification performance when shape features or SBRs were used alone and in combination.

## Statistical analysis

The means of features in the PD and NC groups were tested for significant difference using Welch's t-test. The features were ranked by p-value. Furthermore, receiver operating characteristic (ROC) analysis was performed with the top five features and ML. We used the DeLong test to examine the difference in area under the curves (AUCs) between SBRs alone and shape features alone and SBRs with shape feature. The sensitivity, specificity, positive predictive value (PPV), and negative predictive value (NPV) of each feature were calculated using the optimal cut-off values determined on the basis of ROC analysis. Differences with p-values <0.05 were considered statistically significant.

Statistical calculations were carried out using JMP Pro 12 (SAS, Cary, NC, USA).

## Results

Fig 1 shows typical examples of NC and PD summed SPECT images and image features. Fig 2 shows the comparisons of shape and intensity features between the NC and PD groups. In shape features, area, equivocal diameter, major axis length, perimeter and circularity indicated

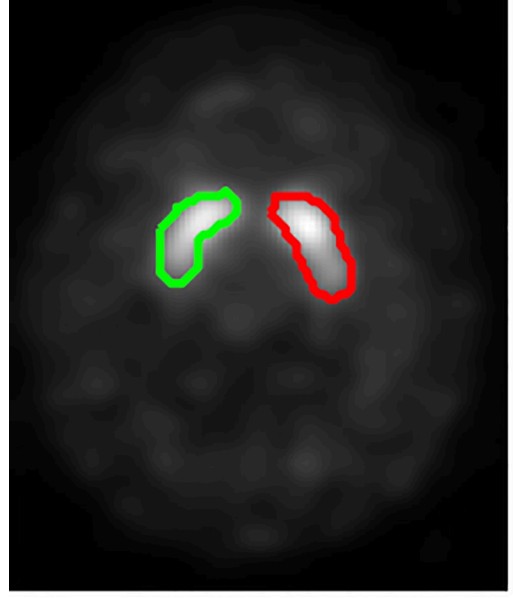
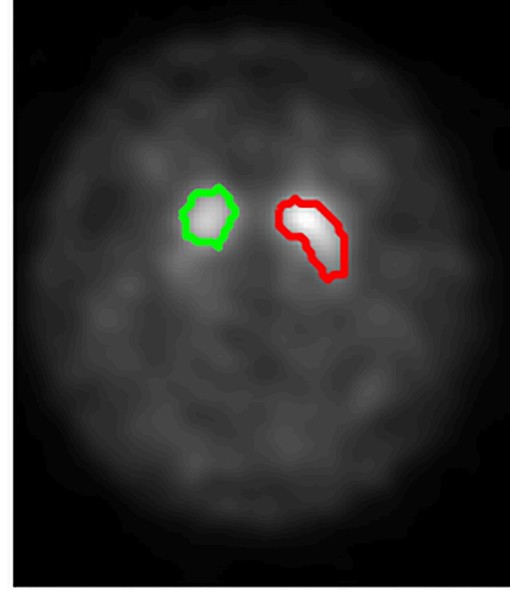

| NC | | | PD | |
| R | L | | R | L |
| 150 | 183 | Area | 84 | 119 |
| 21.2 | 23.6 | EquivDiameter | 11.3 | 18.2 |
| 10.1 | 10.3 | Perimeter | 9.7 | 8.9 |
| 13.8 | 15.3 | MajorAxisLength | 10.3 | 12.3 |
| 49.1 | 52.9 | MinorAxisLength | 32.2 | 42.3 |
| 0.8 | 0.8 | Circularity | 1.0 | 0.8 |

**Fig 1. Example of summed SPECT images and region of interests (ROIs) settings for normal control (left) and Parkinson's disease (right).** ROIs set on right (green line) and left (red line) striata, respectively. Shape features shows under each summed SPECT images.

significant differences (Fig 2A–2E, p < 0.001). The circularity of the PD group was higher than that of the NC group. Minor axis length did not indicate significant difference (Fig 2D, p = 0.1091). In the intensity features, maximum and mean counts indicated significant differences (Fig 2G and 2I, p < 0.001), and the minimum count did not indicate significant difference (Fig 2H, p = 0.5102).

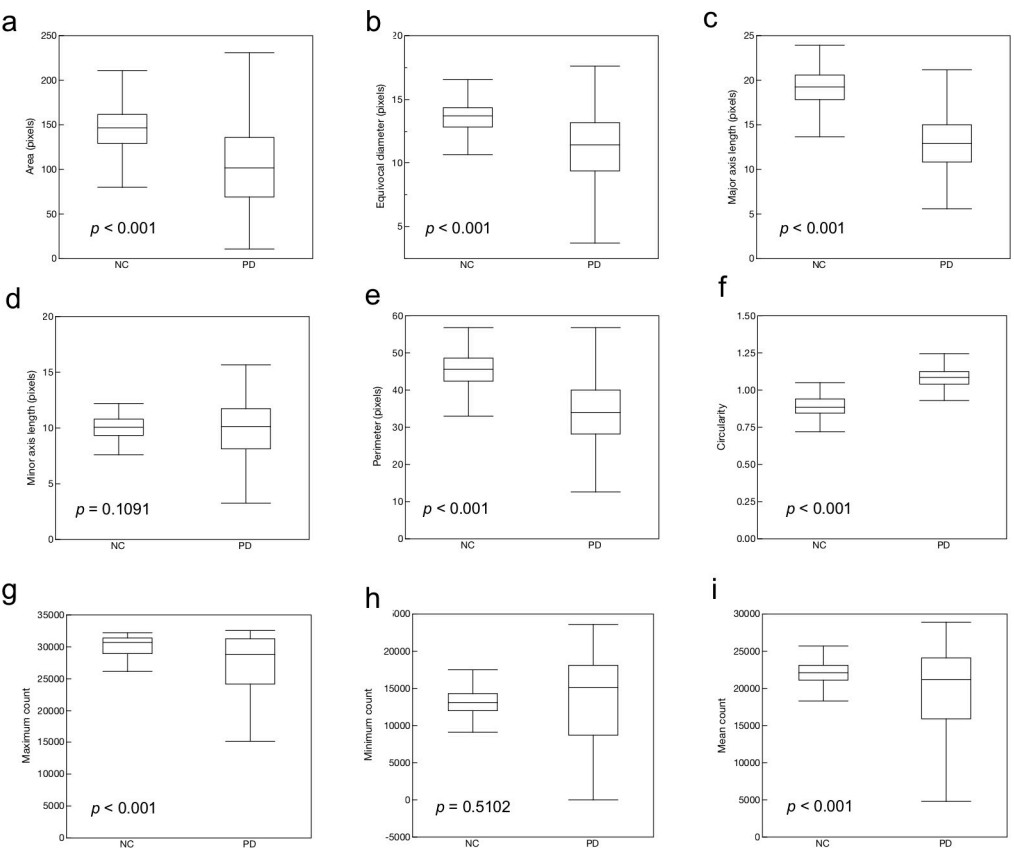

**Fig 2. Comparisons of various features between normal control (NC) and Parkinson's disease (PD) groups.** The features are area (a), equivocal diameter (b), major axis length (c), minor axis length (d), perimeter (e), circularity (f), maximum count (g), minimum count (h), and mean count (i).

Fig 3 shows the comparisons of SBR$_{putamen}$ and SBR$_{caudate}$ between the NC and PD groups. Both SBR$_{putamen}$ and SBR$_{caudate}$ indicated significant differences (p < 0.001). All features ranked in the ascending order of p-values are shown in Table 1. The top five features were SBR$_{putamen}$, circularity, major axis length, SBR$_{caudate}$ and perimeter. The ROC curves for the top five features are shown in Fig 4, and summarized in Table 2. The AUC from the highest to

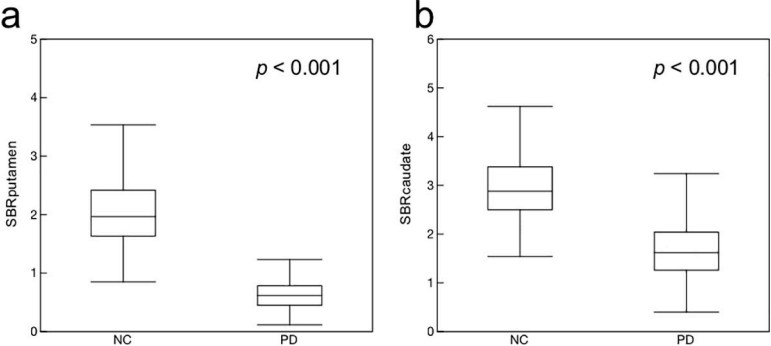

**Fig 3. Comparisons of striatum binding ratios between normal control (NC) and Parkinson's disease (PD) groups.** (a) SBR$_{putamen}$, (b) SBR$_{caudate}$.

**Table 1. Various image features ranked by p-values.**

| Ranking | Features | p values |
|---|---|---|
| 1 | $SBR_{putamen}$ | 5.72E–88 |
| 2 | Circularity | 2.80E–70 |
| 3 | Major axis length | 1.72E–68 |
| 4 | $SBR_{caudate}$ | 2.39E–63 |
| 5 | Perimeter | 2.14E–44 |
| 6 | Area | 2.25E–27 |
| 7 | Equivocal diameter | 7.64E–27 |
| 8 | Maximum count | 2.58E–12 |
| 9 | Mean count | 6.92E–08 |
| 10 | Minor axis length | 1.09E–01 |
| 11 | Minimum count | 5.10E–01 |

*SBR* striatum binding ratio

the lowest were as follows: circularity (0.972), $SBR_{putamen}$ (0.972), major axis length (0.945), $SBR_{caudate}$ (0.928) and perimeter (0.896). Significant differences observed for both $SBR_{putamen}$ vs $SBR_{caudate}$ (p < 0.0001), and $SBR_{putamen}$ vs perimeter (p < 0.0001).

Fig 5 shows ROC curves for ML with circularity or SBRs only and those with the combination. The highest AUC was obtained when the circularity and SBRs were combined (AUC = 0.995), followed by the circularity (AUC = 0.990), and then SBRs (AUC = 0.973). The classification performance was significantly improved by combining SBRs and circularity than by SBRs alone (p = 0.018) shown in Table 3. No significant difference was observed between SBRs

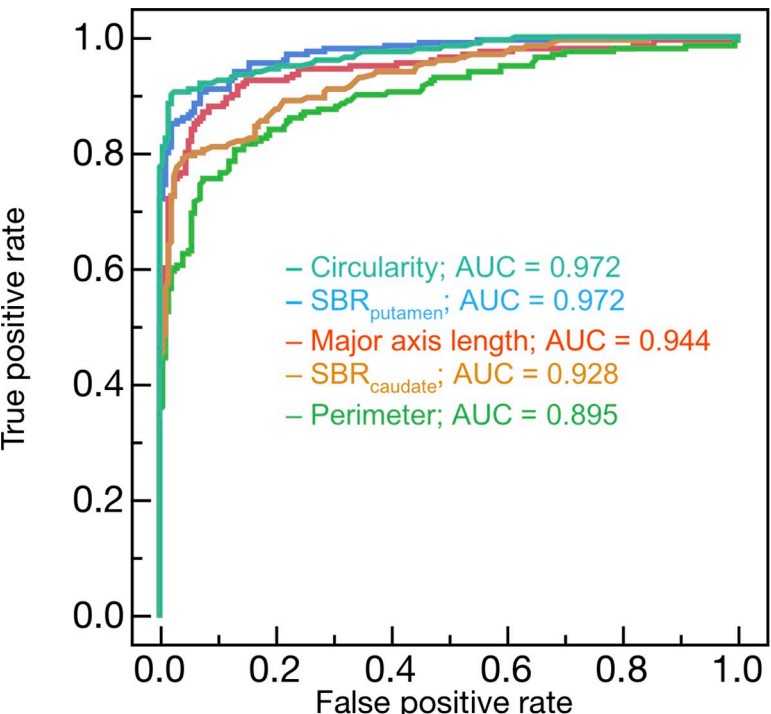

**Fig 4. Receiver operating characteristic curves for the top five features.**

**Table 2. Area under the receiver operating characteristic curve of top five features.**

| Features | AUC | 95% CI | p value (vs $SBR_{putamen}$) |
|---|---|---|---|
| $SBR_{putamen}$ | 0.972 | 0.954–0.984 | NA |
| $SBR_{caudate}$ | 0.928 | 0.900–0.950 | <0.0001 |
| Circularity | 0.972 | 0.955–0.983 | 0.9842 |
| Major axis length | 0.945 | 0.916–0.964 | 0.0394 |
| Perimeter | 0.896 | 0.859–0.924 | <0.0001 |

*SBR* striatum binding ratio, *AUC* area under the curve, *CI* confidence interval, *NA* not applicable

and circularity alone (p = 0.118) and between the combination and circularity alone (p = 0.208). The sensitivity and specificity are summarized in Table 4. Classification accuracy was improved by combining SBRs and circularity than by SBRs alone.

## Discussion

In this study, we evaluated the potential of shape features obtained from DAT-SPECT image to distinguish between the NC and PD groups. The shape features showed high performance equivocal to SBRs.

Obviously, the shape features indicated significant differences between the NC and PD groups, except the minor axis length. In normal cases, the DAT distribution is looks like a comma. On the contrary, in PD cases, DAT distribution has an egg shape. Therefore, the area, perimeter, equivocal diameter, circularity, and major axis lengths are affected by DAT distribution in the striatum. Oliveira et al. reported that the major axis length of the striatal region

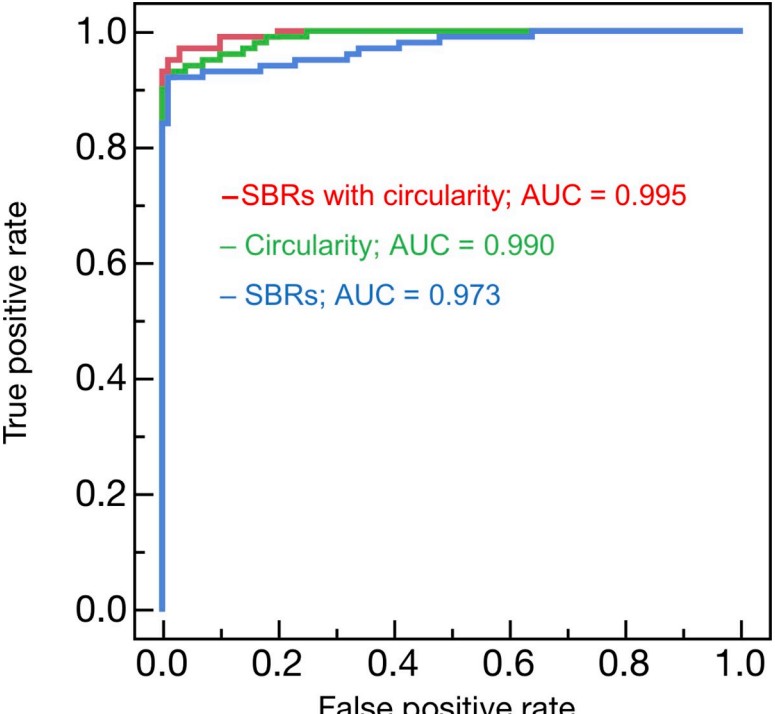

**Fig 5. Receiver operating characteristic curves for the striatum binding ratios (SBRs) alone and circularity alone and in combination.**

**Table 3. Area under the receiver operating characteristic curve of machine learning with several features.**

| Features | AUC | 95% CI | p value (vs SBRs) |
|---|---|---|---|
| SBRs | 0.973 | 0.942–0.987 | NA |
| Circularity | 0.990 | 0.977–0.996 | 0.118 |
| SBRs with Circularity | 0.995 | 0.985–0.998 | 0.018 |

*SBRs* striatum binding ratios, *AUC* area under the curve, *CI* confidence interval, *NA* not applicable

uptake is clinically useful and highly valuable to confirm dopaminergic degeneration as an aid to the diagnosis of Parkinson's disease [13]. However, minor axis length has not affected by comma and/or egg shape.

From the results of ROC analysis, we revealed that the performance of SBRs and shape features are equivalent. Circularity indicated the highest distinguishing performance among the shape features. The reason is that circularity is a mixed index of both area (rank 6th) and perimeter (rank 5th) which have moderate distinguishing performance. In comparison between SBRs, $SBR_{putamen}$ showed high performance. [123]I-FP-CIT decline begins from the putamen in PD, but accumulation is maintained in the caudate of both PD and NC. Therefore, $SBR_{putamen}$ reflected accumulation difference of putamen and showed high performance. Intensity features showed low performance. When using the intensity of the striatum as an index, it should be used as a ratio to the background like SBR. In addition, semi-quantitative evaluation index such as SBR has various calculation methods, it is necessary to compare with these methods.

We compared the classification performance and accuracy of SVM when SBRs were used alone and when circularity was used in combination with SBRs to explore the effectiveness of the shape features in the classification of PD and NC. As a result, the classification performance and accuracy were improved. This result shows the effectiveness of adding a shape feature to SBRs. However, classification performance may be decreased depending on combined shape features. Therefore, the choice of effective shape features is important. We selected shape features for SVM on the basis of p-values by Welch's t-test. In the case of ML using many features, it is necessary to select a method that considered the interaction between the features.

Recently, a study reported the use of texture analysis [23,24]. They reported a number of Haralick textural features in correlation with the clinical measures of UPDRS and disease duration. Further performance improvement could be expected using both shape and textural features together for classification. In future, it would be necessary to apply shape and textural features to ML. Based on the results of this study, shape features will be useful to distinguish between the NC and PD groups by ML. Furthermore, shape features with ML would be useful to distinguish regular PD from atypical PD and/or Parkinsonism.

**Table 4. Classification accuracy of machine learning using SBRs and circularity as a feature.**

| Features | Sensitivity (%) | Specificity (%) |
|---|---|---|
| SBRs | 96.0 | 92.0 |
| Circularity | 97.0 | 93.0 |
| SBRs with Circularity | 98.0 | 95.0 |

*SBRs* striatum binding ratios

This study has some limitations. In this study, we used SPECT images pre-processed by PPMI. These images were reconstructed by a specific reconstruction method and normalized as preprocessing. However, differences in the image reconstruction method and normalization may affect the setting of ROI, and ultimately affect the calculation of the image features. In particular, when the count of the striatum is very low, it is assumed that these influences are large, and there is a possibility that the decreasing classification performance. ROIs for obtaining the shape features were set to the striatum by manual tracing. Thus, the shape features were affected by individual observational differences. To reduce the individual differences, an automatic ROI setting method should be developed. We used only 100 cases of both PD and NC. Therefore, to improve the robustness of the proposed method, future investigations should consider increasing the number of cases.

In conclusion, we found that the shape feature, namely circularity obtained from DAT-SPECT images, could help in distinguishing between NC and PD comparable to SBRs. Furthermore, the classification performance of ML was significantly improved using circularity as a semi-quantitative indicator together. Therefore, circularity can be a useful quantitative index in the diagnostic process of PD and NC.

## Acknowledgments

Data used in this study were obtained from the PPMI database (www.ppmi-info.org/data). PPMI, a public–private partnership, is funded by the Michael J. Fox Foundation for Parkinson's Research and funding partners, including Abbvie, Avid, Biogen, Bristol-Myers Squib, Covance, GE Healthcare, Genentech, GlaxoSmithKline, Eli Lilly & Co, Lundbeck, Merck, Meso Scale Discovery, Pfizer, Piramal, Roche, Servier and UCB.

## Author Contributions

**Conceptualization:** Takuro Shiiba.

**Data curation:** Yuki Arimura, Miku Nagano, Tenma Takahashi.

**Formal analysis:** Takuro Shiiba.

**Funding acquisition:** Takuro Shiiba.

**Investigation:** Takuro Shiiba.

**Methodology:** Takuro Shiiba.

**Project administration:** Takuro Shiiba.

**Supervision:** Takuro Shiiba.

**Writing – original draft:** Takuro Shiiba.

**Writing – review & editing:** Akihiro Takaki.

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
