## [Decision Letter · Decision Letter 0]

3 Oct 2019

PONE-D-19-23329

Improvement of classification performance of Parkinson’s disease using shape features for machine learning on dopamine transporter single photon emission computed tomography

PLOS ONE

Dear Dr Shiiba,

Thank you for submitting your manuscript to PLOS ONE. After careful consideration, we feel that it has merit but does not fully meet PLOS ONE’s publication criteria as it currently stands. Therefore, we invite you to submit a revised version of the manuscript that addresses the points raised during the review process.

We would appreciate receiving your revised manuscript by Nov 17 2019 11:59PM. To enhance the reproducibility of your results, we recommend that if applicable you deposit your laboratory protocols in protocols.io, where a protocol can be assigned its own identifier (DOI) such that it can be cited independently in the future. For instructions see: http://journals.plos.org/plosone/s/submission-guidelines#loc-laboratory-protocols

We look forward to receiving your revised manuscript.

Kind regards,

Jan Kassubek

Academic Editor

PLOS ONE

Journal Requirements:

Additional Editor Comments (if provided):

Although both reviewers saw strengths in your manuscript, both also noted major limitations which should be addressed in a revised version.

Reviewers' comments:

Reviewer's Responses to Questions

**Comments to the Author**

1. Is the manuscript technically sound, and do the data support the conclusions?

Reviewer #1: Partly

Reviewer #2: Partly

2. Has the statistical analysis been performed appropriately and rigorously? 

Reviewer #1: No

Reviewer #2: Yes

3. Have the authors made all data underlying the findings in their manuscript fully available?

Reviewer #1: Yes

Reviewer #2: No

4. Is the manuscript presented in an intelligible fashion and written in standard English?

Reviewer #1: Yes

Reviewer #2: Yes

5. Review Comments to the Author

Reviewer #1: This is an interesting study applying a novel and promising method.

I have only one (yet major) issue:

It is unclear to me, how the boundaries for the shape of the striatum were set. It reads like it was done visually, which is suboptimal, given that the aim of the method is automated quantification, or diagnosis.

There are several other minor issues:

1. Please state in the methods that you downloaded the reconstructed imaging data. The preprocessing steps were performed our of your hands and should be referenced (e.g. in a prior publication, or website reference)

2. How did you determine, what a "burst striatum" constitutes, when you excluded this data?

3. How did you search for the maximum value of the striatal part to determine the medium of the 5 slices that you averaged?

4. line 243 (page 16) : typo "distribution *is* looks like"

5. either use sensitivity/specificity or true positive/negative rate as terminology, not both, which can be confusing

6. To state PPV and NPV in a preselected sample is not meaningful and can be misleading (see consensus statement in https://doi.org/10.1016/j.dadm.2019.01.011, PMID 30984816)

Reviewer #2: Materials and methods

Line 120: please specify which method for image normalization was used and how correct normalization was checked for. In datasets with little striatal activity all normalization methods not using coregistered morphological images for determination of normalization parameters are error-prone and may lead to considerable distortion of images, particularly when the primary reconstructed dataset has unusual starting coordinates as often is the case in parkinsonian patients with stiff neck musculature. As shape feature extraction is particularly prone to bias from distortion due to normalization procedure, requirements on robustness of normalization are high and would need some kind of assessment. One could think of submitting the same dataset to normalization with a couple of differently angulated starting coefficients and measuring reproducibility of shape measures for different extremes, especially for cases where there is little putaminal activity left.

Calculation of image features:

Line 130ff: Are the authors suggesting that for image feature extraction no differentiation between caudate and putaminal activity was done? If so, this would have to be stated explicitly and its implication for disease subentities with predominant caudate pathology discussed. Also, the procedure to determine the „slice with highest striatal activity“ has to be described, as the results will be different if highest activity is measured for caudate, putaminal or global regional activity, and dorsal putaminal slices might not be contributing to summed images if highest activity is in apicoventral caudate. Also, slice thickness of reconstructed images is not reported, but has considerable impact on summed images. Please specify.

Line 133: „regions of interest were set...“. What were the criteria for delineation of striatum? Did the technologist delineate based on morphological images or did he just „paint a contour where activity was“? Were there standardized criteria for contouring besides „ten years of experience?“ - As contouring may have considerable effect on shape feature extraction, as the authors correctly note in the discussion section, it has to be explained how bias by contouring was addressed, especially in „low-count-images“, and Assessment of error propagation should be done for reconstruction-attenuation correction-normalization-contouring-feature extraction.

Results:

Figures show only results of statistical tests. Without image examples, the reader cannot assess the impact of shape feature extraction on tests. Typical images and according shape features should be presented additionally.

Discussion:

The authors should discuss the effect of their specific reconstruction method on the results as compared to standard clinical reconstructed images. On this behalf, it is suggested to repeat their classification methods with reconstructed datasets as present in the database and compare with their standardized reconstructed datasets in order to assess robustness of classification under standard clinical conditions.

6. PLOS authors have the option to publish the peer review history of their article (what does this mean?). If published, this will include your full peer review and any attached files.

Reviewer #1: No

Reviewer #2: No

---

## [Author Response · Author response to Decision Letter 0]

18 Nov 2019

We are grateful to reviewers for the critical comments and useful suggestions that have helped us to improve our manuscript considerably. As indicated in the responses that follow, we have taken all these comments and suggestions into account in the revised version of our manuscript.

Reviewer #1: This is an interesting study applying a novel and promising method.

I have only one (yet major) issue:

It is unclear to me, how the boundaries for the shape of the striatum were set. It reads like it was done visually, which is suboptimal, given that the aim of the method is automated quantification, or diagnosis.

Response

We strongly appreciate the reviewer #1’s comment. A radiological technologist with 10 years of experience in nuclear medicine visually set the boundaries of the striatum. In accordance with reviewer’s comment, we have added the text in the Materials and Methods (Page 10 , Line 146–147): 

“The region where the radioactivity is visually accumulated at the site where the striatum exists anatomically was surrounded.”

We believe that the development of an automatic diagnosis system that takes advantage of shape features can be divided into two parts. One is the extraction of the striatum. The other is the calculation and selection of effective shape features. The purpose of this study is the latter. Therefore, in accordance with reviewer’s comment, we have added the text in the Introduction as follows (Page 6, Line 85–88):

“The Development of an automatic DAT-SPECT diagnosis system that takes advantage of shape features can be divided into two parts. One is the extraction of the striatum. The other is the calculation and selection of effective shape features. We focused on calculation and selection of shape features.”

There are several other minor issues:

1. Please state in the methods that you downloaded the reconstructed imaging data. The preprocessing steps were performed our of your hands and should be referenced (e.g. in a prior publication, or website reference)

Response

In accordance with reviewer’s comment, we have added the text in the Materials and Methods as follows (Page 6, Line 96–99):

“All data used in this study were obtained from the PPMI database (www.ppmi- info.org/data) available on April 3, 2018. The dataset contained all 625 pre-processed 123I-FP-CIT SPECT brain images acquired at the screening stage.”

2. How did you determine, what a "burst striatum" constitutes, when you excluded this data?

Response

It was an expression error. It was not “excluded”, but correctly "was not included". In accordance with reviewer’s comment, we have revised the text in the Materials and Methods from:

“SPECT images of the burst striatum type [9,11,21] were excluded.” 

to

“SPECT images of the burst striatum type [9,11,21] were not included in selected groups.” (Page 7, Line 101–102)

3. How did you search for the maximum value of the striatal part to determine the medium of the 5 slices that you averaged?

Response

As a preliminary experiment, we searched for the maximum value in all slices above the parotid gland. As a result, we found that the maximum value almost exists in either the left or right striatum. In accordance with reviewer’s comment, we have revised the text in the Materials and Methods as follows (Page 9, Line137–142):

“Preliminary experiments showed that the maximum pixel value above the parotid gland was in the left or right striatum. First, the highest pixel value and position of each slice above the parotid gland were searched. Next, a slice with the maximum value of the striatal part was searched.”

4. line 243 (page 16) : typo "distribution *is* looks like"

Response

Thank you for your careful peer review. Our manuscript has been checked natively, how should it be corrected?

5. either use sensitivity/specificity or true positive/negative rate as terminology, not both, which can be confusing

Response

In accordance with reviewer’s comment, in the revised version, we unified the terms (sensitivity and specificity). Vertical and horizontal axes have changed in Figures 3 and 4.

6. To state PPV and NPV in a preselected sample is not meaningful and can be misleading (see consensus statement in https://doi.org/10.1016/j.dadm.2019.01.011, PMID 30984816)

Response

In accordance with reviewer’s comment, we decided to not use PPV and NPV. We have removed PPV and NPV in Table 4, and have revised the text in the Results from:

“The sensitivity, specificity, PPV, and NPV are summarized in Table 4.”

to

“The sensitivity and specificity are summarized in Table 4.” (Page 15, Line 227–228)

We wish to thank the reviewer again for his or her valuable comments.

Reviewer #2: Materials and methods

Line 120: please specify which method for image normalization was used and how correct normalization was checked for. In datasets with little striatal activity all normalization methods not using coregistered morphological images for determination of normalization parameters are error-prone and may lead to considerable distortion of images, particularly when the primary reconstructed dataset has unusual starting coordinates as often is the case in parkinsonian patients with stiff neck musculature. As shape feature extraction is particularly prone to bias from distortion due to normalization procedure, requirements on robustness of normalization are high and would need some kind of assessment. One could think of submitting the same dataset to normalization with a couple of differently angulated starting coefficients and measuring reproducibility of shape measures for different extremes, especially for cases where there is little putaminal activity left.

Response

We strongly appreciate the reviewer ’s comment. PPMI does not disclose details about image normalization. To our knowledge, it was not mentioned in other papers. Therefore, we added following the text to discussion (P19, L283–289).

“In this study, we used SPECT images pre-processed by PPMI. These images were reconstructed by a specific reconstruction method and normalized as pre-processing. However, differences in the image reconstruction method and normalization may affect the setting of ROI, and ultimately affect the calculation of the image features. In particular, when the count of the striatum is very low, it is assumed that these influences are large, and there is a possibility that the decreasing classification performance.”

Calculation of image features:

Line 130ff: Are the authors suggesting that for image feature extraction no differentiation between caudate and putaminal activity was done? If so, this would have to be stated explicitly and its implication for disease subentities with predominant caudate pathology discussed. Also, the procedure to determine the „slice with highest striatal activity“ has to be described, as the results will be different if highest activity is measured for caudate, putaminal or global regional activity, and dorsal putaminal slices might not be contributing to summed images if highest activity is in apicoventral caudate. Also, slice thickness of reconstructed images is not reported, but has considerable impact on summed images. Please specify.

Response 

We searched for the highest pixel value in each slice above the parotid gland and selected the slice with the highest pixel value. As a result, we confirmed that the highest pixel value exists in either the left or right striatum in this database. An image was created by adding the upper and lower slices to the slice. We have added description about image matrix size (Page 9, Line 126–127). 

“The pre-processed images were saved as a DICOM format using 91 × 109 × 91 cubic voxels with 2 mm. “

In accordance with reviewer’s comment, we have revised the text in the Materials and Methods as follows (Page 9, Line 1379–144):

“Preliminary experiments showed that the maximum pixel value above the parotid gland was in the left or right striatum. First, the highest pixel value and position of each slice above the parotid gland were searched. Next, a slice with the maximum value of the striatal part was searched. Then, a summed image was generated from the slice with maximum value and plus or minus two slices from the upper and lower slices (summed range: 1 cm).”

Line 133: „regions of interest were set...“. What were the criteria for delineation of striatum? Did the technologist delineate based on morphological images or did he just „paint a contour where activity was“? Were there standardized criteria for contouring besides „ten years of experience?“ - As contouring may have considerable effect on shape feature extraction, as the authors correctly note in the discussion section, it has to be explained how bias by contouring was addressed, especially in „low-count-images“, and Assessment of error propagation should be done for reconstruction-attenuation correction-normalization-contouring-feature extraction.

Response 

Based on anatomical knowledge, a radiological technologist has set a region of interest in where radioactivity exists visually in the striatum. The summation image is used instead of one slice so that the region of interest setting of the striatum is less biased.

We added following the text to Materials and Methods as following (Page 9, Line 137–139):

 “We thought that the error and bias would increase if the contrast between the striatum and the background was low in a single SPECT image for ROI settings.”

Also, we added following the text to discussion as following (P19, L283–289):

“In this study, we used SPECT images pre-processed by PPMI. These images were reconstructed by a specific reconstruction method and normalized as preprocessing. However, differences in the image reconstruction method and normalization may affect the setting of ROI, and ultimately affect the calculation of the image features. In particular, when the count of the striatum is very low, it is assumed that these influences are large, and there is a possibility that the decreasing classification performance.”

Results:

Figures show only results of statistical tests. Without image examples, the reader cannot assess the impact of shape feature extraction on tests. Typical images and according shape features should be presented additionally.

Response

A typical example of NC and PD images and image features was added as Figure 1. Along with above, we added text to Results as follows:

“Figure 1 shows typical examples of NC and PD summed SPECT images and image features.”

“Fig 1 Example of summed SPECT images and region of interests (ROIs) settings for normal control (left) and Parkinson’s disease (right). ROIs set on right (green line) and left (red line) striata, respectively. Shape features shows under each summed SPECT images.” 

Discussion:

The authors should discuss the effect of their specific reconstruction method on the results as compared to standard clinical reconstructed images. On this behalf, it is suggested to repeat their classification methods with reconstructed datasets as present in the database and compare with their standardized reconstructed datasets in order to assess robustness of classification under standard clinical conditions.

Response

Thank you for your helpful comments. Since we do not have a standard reconstruction processing device, we added the Discussion as follows (P19, L283–289):

 “In this study, we used SPECT images pre-processed by PPMI. These images were reconstructed by a specific reconstruction method and normalized as preprocessing. However, differences in the image reconstruction method and normalization may affect the setting of ROI, and ultimately affect the calculation of the image features. In particular, when the count of the striatum is very low, it is assumed that these influences are large, and there is a possibility that the decreasing classification performance.”

We wish to thank the Reviewer again for his or her valuable comments.

---

## [Decision Letter · Decision Letter 1]

31 Dec 2019

PONE-D-19-23329R1

Improvement of classification performance of Parkinson’s disease using shape features for machine learning on dopamine transporter single photon emission computed tomography

PLOS ONE

Dear Dr Shiiba,

Thank you for submitting your manuscript to PLOS ONE. After careful consideration, we feel that it has merit but does not fully meet PLOS ONE’s publication criteria as it currently stands. Therefore, we invite you to submit a revised version of the manuscript that addresses the points raised during the review process.

We would appreciate receiving your revised manuscript by Feb 14 2020 11:59PM. To enhance the reproducibility of your results, we recommend that if applicable you deposit your laboratory protocols in protocols.io, where a protocol can be assigned its own identifier (DOI) such that it can be cited independently in the future. For instructions see: http://journals.plos.org/plosone/s/submission-guidelines#loc-laboratory-protocols

We look forward to receiving your revised manuscript.

Kind regards,

Jan Kassubek

Academic Editor

PLOS ONE

Additional Editor Comments (if provided):

The reviewers appreciated the detailed revision of the manuscript. However, Reviewer 2 still raises some remaining concerns which might be addressed in a (minor) revision of the manuscript.

Reviewers' comments:

Reviewer's Responses to Questions

**Comments to the Author**

1. If the authors have adequately addressed your comments raised in a previous round of review and you feel that this manuscript is now acceptable for publication, you may indicate that here to bypass the “Comments to the Author” section, enter your conflict of interest statement in the “Confidential to Editor” section, and submit your "Accept" recommendation.

Reviewer #1: All comments have been addressed

Reviewer #2: (No Response)

2. Is the manuscript technically sound, and do the data support the conclusions?

Reviewer #1: Yes

Reviewer #2: Partly

3. Has the statistical analysis been performed appropriately and rigorously? 

Reviewer #1: Yes

Reviewer #2: Yes

4. Have the authors made all data underlying the findings in their manuscript fully available?

Reviewer #1: Yes

Reviewer #2: No

5. Is the manuscript presented in an intelligible fashion and written in standard English?

Reviewer #1: Yes

Reviewer #2: Yes

6. Review Comments to the Author

Reviewer #1: (No Response)

Reviewer #2: The comments raised are partially addressed. While the revised manuscript now comments on preprocessing by PPMI, the methods section still comments confusingly on processing of SPECT raw data, which is not intelligible by an average reader not accustomed to the PPMI database. In fact, the reviewer had to gain access to the PPMI database and check himself on the data to fully understand what the authors were doing. Please consider the following points:

line 130ff: the authors still do not specify if striatal activity was differentiating caudate activity from putaminal activity. It seems they just differentiated left striatal activity from right striatal activity. They might insert a line "left and right striatal ROIs were covering and including all activity visualised in putamen and caudate"

Line 113 typo PPIM instead of PPMI

Line 114: "SPECT images and SBRs were downloaded from the PPMI website" - please change to "Preprocessed SPECT images and SBRs were downloaded from the PPMI website".

Line 114 ff: "SPECT imaging was acquired at each imaging centers as per the PPMI imaging protocol..." The current wording suggests that this part of processing had been done by the authors. In fact the whole paragraph following is plagiating the PPMI methods section where the preprocessing steps performed by PPMI are described, which is misleading the reader. Please consider the following wording:

"As by PPMI documentation, preprocessing steps were performed at the Institute for Neurodegenerative Disorders (IND, New Haven, CT) and included the following steps: SPECT imaging and reconstruction: SPECT imaging "was acquired at each imaging centers as per the PPMI imaging protocol [..., up to line 127]. (line 128 insert:) The calculation method of SBR as performed at the IND was as follows: the transaxial slice with...

line 146: " The region where the radioactivity is

visually accumulated at the site where the striatum exists anatomically was surrounded" - please replace "surrounded" by "manually delineated"

Additional comment: reviewer 1 asked for definition of "burst striatum" which is not given in the revised document. Please provide.

7. PLOS authors have the option to publish the peer review history of their article (what does this mean?). If published, this will include your full peer review and any attached files.

Reviewer #1: Yes: Thilo van Eimeren

Reviewer #2: No

---

## [Author Response · Author response to Decision Letter 1]

7 Jan 2020

We are grateful to reviewers for the critical comments and useful suggestions that have helped us to improve our manuscript considerably. As indicated in the responses that follow, we have taken all these comments and suggestions into account in the revised version of our manuscript.

line 130ff: the authors still do not specify if striatal activity was differentiating caudate activity from putaminal activity. It seems they just differentiated left striatal activity from right striatal activity. They might insert a line "left and right striatal ROIs were covering and including all activity visualised in putamen and caudate"

Response

We strongly appreciate the reviewer’s comment. In accordance with reviewer’s comment, we have changed the text in the Materials and Methods (P10L134) as from:

“Regions of interests (ROIs) were placed on the left and right caudate and putamen (target region) and the occipital cortex (reference region).”

to

“Regions of interests (ROIs) were placed on the left and right (target region) and the occipital cortex (reference region).”

Line 113 typo PPIM instead of PPMI

Response

We strongly appreciate the reviewer’s comment. In accordance with reviewer’s comment, we have revised from “PPIM” to “PPMI”(P8L114). 

Line 114: "SPECT images and SBRs were downloaded from the PPMI website" - please change to "Preprocessed SPECT images and SBRs were downloaded from the PPMI website".

Response

We strongly appreciate the reviewer’s comment. In accordance with reviewer’s comment, we have changed the text in the Materials and Methods (P8L115) as from:

“SPECT images and SBRs were downloaded from the PPMI website.”

to

“Preprocessed SPECT images and SBRswere downloaded from the PPMI website.”

Line 114 ff: "SPECT imaging was acquired at each imaging centers as per the PPMI imaging protocol..." The current wording suggests that this part of processing had been done by the authors. In fact the whole paragraph following is plagiating the PPMI methods section where the preprocessing steps performed by PPMI are described, which is misleading the reader. Please consider the following wording:

"As by PPMI documentation, preprocessing steps were performed at the Institute for Neurodegenerative Disorders (IND, New Haven, CT) and included the following steps: SPECT imaging and reconstruction: SPECT imaging "was acquired at each imaging centers as per the PPMI imaging protocol [..., up to line 127]. (line 128 insert:) The calculation method of SBR as performed at the IND was as follows: the transaxial slice with...

Response

We strongly appreciate the reviewer’s comment. In accordance with reviewer’s comment, we have added the text in the Materials and Methods (P8L116) as follows:

As by PPMI documentation, preprocessing steps were performed at the Institute for Neurodegenerative Disorders (IND, New Haven, CT) and included the following steps: SPECT imaging and reconstruction:

Also, we have changed the text (P9L132) as following:

The calculation method of SBR as performed at the IND was as follows: the transaxial slice with the highest striatal uptake was identified, and the eight hottest striatal slices around it were averaged to generate a single slice image.

line 146: " The region where the radioactivity is visually accumulated at the site where the striatum exists anatomically was surrounded" - please replace "surrounded" by "manually delineated"

Response

We strongly appreciate the reviewer’s comment. In accordance with reviewer’s comment, we have changed the text in the Materials and Methods (P10L152) as from:

“The region where the radioactivity is visually accumulated at the site where the striatum exists anatomically was surrounded.”

to

“The region where the radioactivity is visually accumulated at the site where the striatum exists anatomically was manually delineated.”

Additional comment: reviewer 1 asked for definition of "burst striatum" which is not given in the revised document.4 Please provide.

Response

We strongly appreciate the reviewer’s comment. In accordance with reviewer’s comment, we have added the text in the Materials and Methods (P7L105) as following:

The burst striatum type is severe bilateral reduction with almost no uptake in either the putamen or caudate[7].

We wish to thank the Reviewer again for his or her valuable comments.

---

## [Decision Letter · Decision Letter 2]

13 Jan 2020

Improvement of classification performance of Parkinson’s disease using shape features for machine learning on dopamine transporter single photon emission computed tomography

PONE-D-19-23329R2

Dear Dr. Shiiba,

We are pleased to inform you that your manuscript has been judged scientifically suitable for publication and will be formally accepted for publication once it complies with all outstanding technical requirements.

With kind regards,

Jan Kassubek

Academic Editor

PLOS ONE

Additional Editor Comments (optional):

All reviewers´comments have been appropriately addressed.

Reviewers' comments:

Reviewer's Responses to Questions

**Comments to the Author**

1. If the authors have adequately addressed your comments raised in a previous round of review and you feel that this manuscript is now acceptable for publication, you may indicate that here to bypass the “Comments to the Author” section, enter your conflict of interest statement in the “Confidential to Editor” section, and submit your "Accept" recommendation.

Reviewer #2: All comments have been addressed

2. Is the manuscript technically sound, and do the data support the conclusions?

Reviewer #2: Yes

3. Has the statistical analysis been performed appropriately and rigorously? 

Reviewer #2: Yes

4. Have the authors made all data underlying the findings in their manuscript fully available?

Reviewer #2: Yes

5. Is the manuscript presented in an intelligible fashion and written in standard English?

Reviewer #2: Yes

6. Review Comments to the Author

Reviewer #2: The authors have addressed all comments appropriately. There is only one mishap in the revised manuscript, where at page 9, Line 13 comments and original text have been mixed and the manuscript now reads "Regions of interests (ROIs) were placed on the left and right striatal ROIs were covering and including all activity visualised in putamen and caudate (target region), and the occipital cortex (reference region).", while it should read "Regions of interests (ROIs) were covering and including all activity visualised in putamen and caudate (target region), and the occipital cortex (reference region)."

7. PLOS authors have the option to publish the peer review history of their article (what does this mean?). If published, this will include your full peer review and any attached files.

Reviewer #2: Yes: Freimut D. Juengling

---

## [Editor Report · Acceptance letter]

16 Jan 2020

PONE-D-19-23329R2 

Improvement of classification performance of Parkinson’s disease using shape features for machine learning on dopamine transporter single photon emission computed tomography 

Dear Dr. Shiiba:

I am pleased to inform you that your manuscript has been deemed suitable for publication in PLOS ONE. Congratulations! Your manuscript is now with our production department. 

With kind regards,

on behalf of

Prof. Dr. Jan Kassubek 

Academic Editor

PLOS ONE